# Unraveling the role of host kinase PIM1 in *Toxoplasma gondii* infection: Implications for therapies

Lijie Pan[1ʘ], Fan Zhang[1ʘ], Yun Yang[1ʘ], Yue Sun[1,2ʘ], Lingling Song[1], Famin Zhang[1], Jinjin Zhu[1], Yang Wang[1], Chong Wang[1], Qingli Luo[1], Wen Zhang[3*], Li Yu[1*], Yuanyuan Cao[1*]

**1** Department of Microbiology and Parasitology, Anhui Province Laboratory of Zoonoses, School of Basic Medical Sciences, Anhui Medical University, Hefei, Anhui, China, **2** Clinical College of Anhui Medical University, Hefei, Anhui, China, **3** Hefei First People's Hospital, the Third Affiliated Hospital of Anhui Medical University, Hefei, Anhui, China

ʘ These authors contributed equally to this work.
* vickycao1989@126.com (YYC); lilyyu33@126.com (LY); zw112200@126.com (WZ)

## Abstract

*Toxoplasma gondii*, a widespread intracellular protozoan parasite, infects a significant portion of the global human population. Restricted to an acute-infection model, this study elucidates the role of the host protein PIM1, a serine/threonine protein kinase, in facilitating *T. gondii* proliferation and its potential as a therapeutic target. Employing both *in vitro* and *in vivo* models, we establish that PIM1 enhances the intracellular proliferation of *T. gondii* by suppressing host cell apoptosis. Our findings underscore the necessity of PIM1's kinase activity in this process, as evidenced by the significant reduction in *T. gondii* proliferation upon treatment with either a kinase-dead PIM1 mutant or the PIM1 inhibitor AZD1208. In murine models, AZD1208 treatment resulted in decreased *T. gondii* load and elevated pro-apoptotic markers in tissues, indicating that PIM1 inhibition bolsters the host's immune response against the parasite. Since the role of PIM1 in chronic infection remains unexplored, follow-up studies using chronic models are essential. Collectively, our findings illuminate host–parasite interplay during acute toxoplasmosis and position PIM1 as a promising target for anti-*T. gondii* therapeutics.

## Author summary

PIM1 functions as a regulator that suppresses apoptosis and, consequently, fosters the proliferation of *T. gondii*. This promotion of *T. gondii* proliferation by PIM1 is contingent upon its kinase activity, and the inhibition of PIM1 has been shown to induce increased apoptosis in host cells across both *in vitro* and *in vivo* studies. The potential therapeutic application of AZD1208, a PIM1 small molecule

**Data availability statement:** All relevant data are within the manuscript and its Supporting Information files.

**Funding:** This work was funded by the National Natural Science Foundation of China (grant number 81802003) to Y.Y.C., the Scientific Research Foundation of the Education Department of Anhui Province (grant number 2022AH050751) to Y.Y.C., the Anhui Medical University Doctoral Scientific Research Foundation (grant number xj201725) to Y.Y.C., the National Key Laboratory of Pathogen Microbiology Biosafety Open Fund Project (grant number SKLPBS2456) to Y.Y.C., and the Anhui Provincial Key Laboratory of Pathogen Biology Open Fund Project (grant number BY-2022Z04) to W.Z. The funders had no role in study design, data collection and analysis, decision to publish, or preparation of the manuscript.

**Competing interests:** The authors have declared that no competing interests exist.

inhibitor, in treating toxoplasmosis has been brought to light by these findings. AZD1208 has demonstrated the ability to eliminate *T. gondii*-infected cells and decrease the overall parasite load by inhibiting PIM1. Collectively, the research emphasizes the role of PIM1 in the proliferation of *T. gondii* during infection and highlights the therapeutic potential of PIM1 inhibition in the fight against toxoplasmosis. The study's findings propose that targeting PIM1 may offer a viable strategy for the development of innovative treatments aimed at this intracellular parasite.

## Introduction

*Toxoplasma gondii*, a zoonotic intracellular protozoan parasite, is renowned for its ability to invade and infect virtually a wide range of nucleated cells within its host. The transmission of *T. gondii* to humans predominantly occurs through the consumption of undercooked meat harboring viable tissue cysts, as well as through the ingestion of food and water contaminated with *T. gondii* oocysts. Globally, nearly all warm-blooded vertebrates, including humans, act as intermediate hosts, with cats and other members of the Felidae family serving as definitive hosts [1,2]. A systematic review and prevalence snapshot investigation revealed that the global mean seroprevalence rate of *T. gondii* infection stands at 25.7% [3]. Toxoplasmosis, while often asymptomatic in healthy individuals, poses significant risks to pregnant women and immunocompromised individuals. It is estimated to cause substantial economic losses in China, amounting to approximately USD 700 million annually [4]. This includes treatment costs, productivity losses, expenses for animal vaccination, and costs associated with the development of new drugs. In early pregnancy, infection can lead to miscarriage, and congenitally infected infants may suffer from severe conditions such as hydrocephalus, severe ocular infections, and brain tissue abnormalities at birth or during early infancy [5,6]. For those with weakened immune systems, it can trigger central nervous system infections, leading to life-threatening diseases like encephalitis and meningitis. Additionally, it can result in ocular infections that impair vision and systemic infections [1]. Therefore, the harm of toxoplasmosis should not be overlooked, especially for vulnerable groups such as pregnant women, cancer patients, and individuals with HIV infection who are under immunosuppression. Currently, the therapeutic regimen for toxoplasmosis primarily involves the combination of pyrimethamine and sulfadiazine [7,8]. Despite the established efficacy of this regimen, it is recognized that these treatments fall short in completely eradicating tissue cysts, leading to the requirement for extended treatment durations in immunosuppressed individuals. Throughout this extended course, the regimen carries notable toxicities—leukopenia, thrombocytopenia, cutaneous rash, and fever [9,10]—and, although clinically significant resistance remains rare, laboratory selection of dihydrofolate-reductase or -synthase mutations that raise the MIC has been readily achieved [11], underscoring the urgent need for safer, more effective anti-*T. gondii* drugs and vaccines.

The genus *Toxoplasma* includes only one species, *Toxoplasma gondii*. *T. gondii* is classified into type I strains (such as RH, GT1, etc.), type II strains (such as PRU, ME49, etc.), and type III strains (such as VEG, etc.). Different genotypes of *T. gondii* possess distinct genetic backgrounds, which significantly influence their pathogenicity to hosts and their ability to form cysts within the host. Type I strains exhibit the highest virulence, while Type II and III strains show relatively lower pathogenicity. Despite their lower pathogenicity, Type II and III strains are more prone to cyst formation within the host and may possess a higher cyst-forming capacity compared to Type I strains [12–14]. However, recent studies have revealed a much more complex population structure of *T. gondii* than the three classical clonal lineages. Advances in high-throughput sequencing and population genetics have identified many additional genotypes and recombinant strains with unique virulence characteristics and geographic distributions. Globally, non-archetypal and recombinant lineages have been reported in South America, Africa, and Asia, where the genetic diversity of *T. gondii* genotypes is particularly high [15,16]. In Asia, a unique lineage known as the Chinese 1 genotype has been identified as a predominant strain in certain regions of China. The Chinese 1 genotype exhibits intermediate virulence compared to the classical lineages and shows significant regional adaptation, highlighting the genetic plasticity of *T. gondii* populations [17,18]. Understanding this genetic diversity is crucial, as it plays a significant role in the parasite's ability to adapt to different hosts and environmental conditions, as well as in the clinical manifestations of toxoplasmosis. The genetic diversity of *T. gondii* is essential for its adaptation to different hosts and environments. Different genotypes may possess distinct virulence characteristics, which can influence the clinical outcomes of toxoplasmosis in humans and animals. Understanding this genetic diversity is vital for developing effective diagnostic tools, therapeutic strategies, and public health interventions for toxoplasmosis.

Infection by *T. gondii* prompts the host's immune system to utilize a variety of pattern recognition receptors (PRRs) to detect pathogen-associated molecular patterns (PAMPs) specific to *T. gondii* [19–24]. This recognition process initiates the production of inflammatory cytokines and chemokines, which are essential for the host's defense mechanisms. Notably, the interleukin (IL)-12/interferon (IFN)-γ axis is particularly crucial in orchestrating the host's response to *T. gondii* infection [25,26]. Moreover, apoptosis serves as a critical host mechanism for eliminating intracellular pathogens. Initiation of apoptosis, particularly through the intrinsic pathway, involves the activation and oligomerization of pro-apoptotic Bcl-2 family proteins, such as Bax and Bak. This process leads to an increased permeability of the mitochondrial outer membrane, facilitating the release of cytochrome c into the cytosol. The cytosolic cytochrome c then promotes the formation of the apoptosome, which activates caspase-9. Subsequently, caspase-9 triggers the activation of downstream effector caspases, including caspase-3. This cascade of events culminates in nuclear condensation, DNA fragmentation, alterations in cell membrane composition, and the formation of apoptotic bodies, ultimately resulting in cell death [27,28]. Extensive research has established that *T. gondii* can suppress host cell apoptosis, targeting both the mitochondrial and death receptor pathways, which is vital for the parasite's survival and replication within host cells [29–35].

Despite extensive research describing the anti-apoptotic effects of *T. gondii* in human cells, the parasite factors that trigger this response and the host proteins that may play a role in this process remain elusive. Some studies have explored host dependency factors in *T. gondii*-infected cells, including certain amino acids, host proteins, and miRNAs [36–38]. Among these, we have taken note of a study that utilized whole-genome CRISPR screening to identify human host factors required for *T. gondii* ME49 strain infection, with particular interest in the proviral integration of Moloney murine leukemia virus 1 protein kinase (PIM1) [39]. PIM1 is a serine/threonine protein kinase that belongs to the PIM kinase family which plays a significant role in cellular proliferation, differentiation, and survival, including anti-apoptotic functions [40,41]. In addition, it is associated with the development of various types of cancer [42]. Given its role in cancer, PIM1 is considered a potential target for anticancer drug development [43–45]. Researchers are developing small molecule inhibitors that can suppress the activity of PIM1, such as AZD1208 [46–48]. However, the role of PIM1 *in vivo* during *T. gondii* infection and the treatment of toxoplasmosis by AZD1208 have been rarely studied.

Our central hypothesis is that PIM1, a serine/threonine protein kinase, plays a vital role in promoting the proliferation of *T. gondii* by inhibiting intrinsic apoptosis in infected host cells. We propose that targeting PIM1 with small molecule

inhibitors, such as AZD1208, could potentially reduce the *T. gondii* load in infected hosts. To validate this hypothesis and explore its implications for toxoplasmosis therapy, our study goals are as follows: 1. To delineate the role of PIM1 in *T. gondii* proliferation using both in vitro and *in vivo* models; 2. To ascertain the necessity of PIM1's kinase activity for its function in *T. gondii* proliferation; 3. To explore the molecular mechanisms by which PIM1 mediates the anti-apoptotic effect in *T. gondii*-infected cells; 4. To evaluate the efficacy of PIM1 inhibition in reducing *T. gondii* burden *in vivo* and to understand its impact on host immune responses. By achieving these study goals, we aim to enhance the understanding of the host-parasite interactions in *T. gondii* infection and to identify novel therapeutic strategies for combating toxoplasmosis.

## Materials and methods

### Ethics statement

Animal experiments were approved by the Animal Experimental Ethics Committee of Anhui Medical University (Permit No.: 20180243) and conducted in strict accordance with the institution's Guidelines for the Care and Use of Research Animals. Mice were housed in a controlled environment with regulated temperature and humidity, subjected to a 12-hour light/dark cycle, and provided with unlimited access to food and filtered water in standard enclosures. All efforts were made to minimize animal suffering, and humane endpoints were established, including weight loss exceeding 20% of initial body weight or signs of severe distress. Euthanasia was performed using carbon dioxide ($CO_2$) asphyxiation, and all personnel involved in animal handling and euthanasia were trained and certified to ensure the procedures were conducted in a professional and humane manner. The study was designed to ensure that the number of animals used was the minimum required to achieve scientifically valid results, as determined by a power analysis prior to the start of the experiments. All data were collected and analyzed anonymously to ensure that the animals' identities were not linked to the collected data.

### Cell and parasite culture

African Green Monkey Kidney (Vero) cells, Henrietta Lacks (HeLa) cells, Human Embryonic Kidney (HEK 293T) cells, and Mouse Leukemia Monocyte Macrophage (Raw264.7) cells were obtained from our laboratory which was stored in liquid nitrogen tanks. Cells were cultured in Dulbecco's Modified Eagle Medium (DMEM, Viva Cell), supplemented with 10% fetal bovine serum (FBS, NEWZERUM), 100 µg/ml penicillin and 100 µg/ml streptomycin (Sigma, USA). Cultures were maintained in an incubator at 37 °C with 5% $CO_2$. The *T. gondii* ME49 strain was cultured in Vero cells using DMEM containing 2% FBS. Lysates from *T. gondii*-infected Vero cells were purified by sequential centrifugation at $500 \times g$ for 5 minutes to collect the supernatant, followed by a second centrifugation at $2200 \times g$ for 5 minutes to clear the supernatant. The *T. gondii* tachyzoites were resuspended in DMEM with 2% FBS and enumerated before infecting the cells at a multiplicity of infection (MOI) of 3 for 24 hours. The counting method is as follows: mix the parasite suspension 1:1 with 0.4% trypan blue; dead tachyzoites stain blue, live ones remain unstained. Load 10 µL into a Neubauer chamber and count the four large corner squares. Apply the formula: Tachyzoites/mL = (total live parasites in four large squares/ 4) × $10^4$ × dilution factor.

### RNA/DNA extraction and quantitative real-time PCR analysis

Total RNA was extracted from cells and mouse spleens using Trizol Reagent (Invitrogen, USA). This RNA was reverse transcribed using the EvoM-MLV RT Kit with gDNA Clean for qPCR (AG, China), following the manufacturer's protocol. Genomic DNA was extracted from the same samples using the SteadyPure Universal Genomic DNA Extraction Kit (AG, China). For tissues, prior to the extraction of genomic DNA or RNA, a high-throughput tissue homogenizer should be used for processing. The specific method involves adding 100 mg of tissue to the corresponding lysis buffer, followed by subjecting the lysate to short bursts of sonication with an ultrasonic cell disruptor (3–5 s pulses separated by 10 s intervals, repeated 3–5 times). qRT-PCR was performed using the SYBR Green Premix Pro Taq HS qPCR Kit (AG, China) on a LightCycler 96 fluorescence quantitative PCR instrument (Roche, Switzerland). Primer sequences are shown in a table within the Supplementary File.

Relative cDNA levels were determined using the $2^{-\Delta\Delta CT}$ method, with expression values normalized to the housekeeping gene GAPDH. Each sample was analyzed in triplicate and each experiment was repeated a minimum of three times.

## Protein isolation and western blot analysis

$5 \times 10^5$ corresponding cells were seeded into each well of a 12-well plate. After the cells reached 50% confluence, they were infected at MOI = 3 for 24 hours before sample collection. Cells were washed with PBS and lysed in a solution containing cold RIPA Lysis buffer (Beyotime, China), 1% Phenylmethanesulfonyl fluoride (PMSF, Beyotime), and Protease Inhibitor Cocktail (MCE, USA). 50 mg of spleen tissues were homogenized in the same RIPA lysis buffer and incubated on ice for 10 min. The lysate was then subjected to a high-throughput tissue homogenizer (3–5 s pulses separated by 10 s intervals, repeated 3–5 times), followed by centrifugation at $12,000 \times g$ for 10 min in a 4 °C pre-cooled centrifuge. The supernatant was collected as the protein fraction for downstream analysis. Protein samples were mixed with $5 \times$ SDS loading buffer and boiled for 5 minutes. Lysates were resolved on a 15% SDS-PAGE gel and transferred to a PVDF membrane (Billerica, USA). Membranes were blocked with 5% nonfat skim milk in TBST (0.1% Tween 20) for 1.5 hours at room temperature, then incubated with primary antibodies overnight at 4°C, followed by incubation with horseradish peroxidase (HRP)-conjugated secondary antibodies for 1 hour at room temperature. Protein bands were visualized using the ChemiDoc Chemiluminescence Platform (Bio-Rad, USA) and Sparkjade ECL Super (Sparkjade, Shandong, China). Primary antibodies were used at the following dilutions: anti-β-Actin (Proteintech, China) 1:8000, α-Tubulin (Proteintech, China) 1:3000, Bcl-2 (Proteintech, China) 1:2000, Bax (Proteintech, China) 1:2000, Caspase-3 (Proteintech, China) 1:2000, Cleaved-Caspase-3 (CST, USA) 1:1000, HRP-conjugated Goat Anti-Rabbit IgG(H + L) (Proteintech, China) 1:5000, HRP-conjugated Goat Anti-Mouse IgG(H + L) (Proteintech, China) 1:5000.

## Giemsa staining

$2 \times 10^5$ WT or PIM1-KO HeLa cells were cultured on glass coverslips in 12-well plates and infected with *T. gondii* ME49 for 24 hours. Following incubation, the medium was discarded, and the coverslips were washed three times with cold PBS, fixed with 4% paraformaldehyde for 30 minutes, and then washed again with cold PBS. The cells were stained with Giemsa staining solution (Beyotime, China) for 30 minutes and gently rinsed. After air-drying, the coverslips were mounted onto slides using neutral resin for microscopic examination. PVs were counted 100 randomly chosen, non-overlapping fields per coverslip under $100 \times$ oil-immersion objective. PVs were identified as intracellular vacuoles containing ≥1 tachyzoite. Fields were selected by systematic random sampling and every 10th field was imaged and analyzed. Counts were performed by two blinded observers.

## TUNEL apoptosis staining

$2 \times 10^5$ WT or PIM1-KO HeLa cells were cultured on glass coverslips in 12-well plates and infected with *T. gondii* ME49 for 24 hours. Cells were processed according to the immunofluorescence protocol. Briefly, cells were fixed with 4% paraformaldehyde for 20 minutes, permeabilized with 0.02% Triton X-100 for 20 minutes, and blocked with 3% BSA solution for 30 minutes. The cells were then incubated with the primary antibody against GAP45 overnight, followed by incubation with the secondary antibody. Subsequently, cells were subjected to the TUNEL Apoptosis Detection Kit (Beyotime, China), with a final step of counterstaining with Antifade Mounting Medium with DAPI (Beyotime, China). The images were captured using a Zeiss LSM 980 with Airyscan 2 super-resolution confocal microscope.

## Mitochondrial isolation

Mitochondria and cytosol were fractionated from $2 \times 10^7$ cells with the Beyotime Mitochondria Isolation Kit (China). Cells were washed with ice-cold PBS, resuspended in 1 mL ice-cold mitochondrial isolation buffer (1 mM PMSF) and kept on

ice for 10 min. The suspension was homogenized with a glass homogenizer (10–30 strokes; ≥ 50% membrane breakage confirmed by trypan-blue staining) and subjected to sequential centrifugation: 600 × g, 10 min, 4 °C to remove nuclei and unbroken cells; the resulting supernatant was then centrifuged at 11,000 × g, 10 min, 4 °C. The high-speed pellet (mitochondrial fraction) was either resuspended in 150 μL mitochondrial storage buffer for functional assays or in 150 μL lysis buffer (1 mM PMSF) for mitochondrial protein extraction. The high-speed supernatant (cytosolic fraction) was further clarified at 12, 000 × g, 10 min, 4 °C, and the final supernatant was collected as the cytosolic extract.

## Animal model

Female C57BL/6J mice (5 weeks old) were purchased from Beijing Vital River Laboratory Animal Technology Co., Ltd. and divided into six groups (n = 5/group):

Uninfected + DMSO; Uninfected + AZD1208 (30 mg/kg/day, MCE, USA); Uninfected + Pyrimethamine (30 mg/kg/day, Sigma, USA); ME49 infected + DMSO; ME49 infected + AZD1208 (30 mg/kg/day, MCE, USA); ME49 infected + Pyrimethamine (30 mg/kg/day, Sigma, USA). On day 0, mice in groups 4–6 received an intraperitoneal injection of $1 \times 10^5$ *T. gondii* ME49 tachyzoites; groups 1–3 remained uninfected. From day 1 to day 7, each group received its respective treatment intraperitoneally once daily. 12 hours after the final treatment on Day 7, spleens and livers were collected for downstream analyses. All procedures adhered to internationally accepted guidelines for the ethical treatment of laboratory animals and were approved by the Institutional Animal Care and Use Committee of Anhui Medical University.

## Statistical analysis

The data presented are the averages from three independent experiments. Graphs display the mean ± SD (n = 3) from these experiments. Normality was verified with the Shapiro–Wilk test before selecting the parametric test. Figs 1K and 3E: two-way ANOVA followed by Tukey's multiple-comparisons test. Figs 2A, 4B–4F and 5A–5D: one-way ANOVA followed by Tukey's multiple-comparisons test. Other two-group comparisons were analysed with two-tailed Student's t-test (or Mann-Whitney U test when normality was not met).

## Results

### PIM1 promotes intracellular *T. gondii* proliferation

As previously mentioned, studies utilizing whole-genome CRISPR screening have identified PIM1 as a crucial host dependency factor for the replication of *T. gondii* ME49 strain [39]. In our preliminary research, we aimed to ascertain the regulatory function of PIM1 kinase as a putative host factor in modulating the proliferation of type II ME49 strains of *T. gondii*. To achieve this, we utilized a combination of quantitative real-time polymerase chain reaction (qRT-PCR) and western blot (WB) analyses to meticulously examine the impact of PIM1 on the proliferation patterns of this specific *T. gondii* strain across a panel of genetically engineered cell lines. Specifically, we compared the parasite burden of *T. gondii* in Raw264.7 cells with stable PIM1 knockdown, PIM1-null HeLa cells and 293T cells with transient PIM1 transfection, juxtaposed against their corresponding wild-type and control cell lines. Our findings revealed that in the reduced or absent expression of PIM1, there was a significant downregulation of the mRNA levels of the *T. gondii*-specific gene internal transcribed spacer-1 (TgITS-1) (Fig 1A and 1B). Conversely, overexpression of PIM1 was associated with an increase in TgITS-1 mRNA levels (Fig 1C). Additionally, we scrutinized the expression of the *T. gondii* profilin protein. Suppression or deficiency of PIM1 expression resulted in a diminished profilin protein level (Fig 1D and 1E), while overexpression of PIM1 also led to an elevation in profilin protein levels (Fig 1F). The validation of PIM1 expression levels in the genetically modified cell lines was confirmed through WB (Fig 1G-1I). In addition to the ME49 strain, we also examined the effects of PIM1 on the proliferation of the classic Type I RH strain and the Chinese-specific *T. gondii* Chinese 1 genotype Wh6 strain (abbreviated as TgCtWh6). As shown in S1 Fig, deficiency of PIM1 expression resulted in a diminished profilin protein

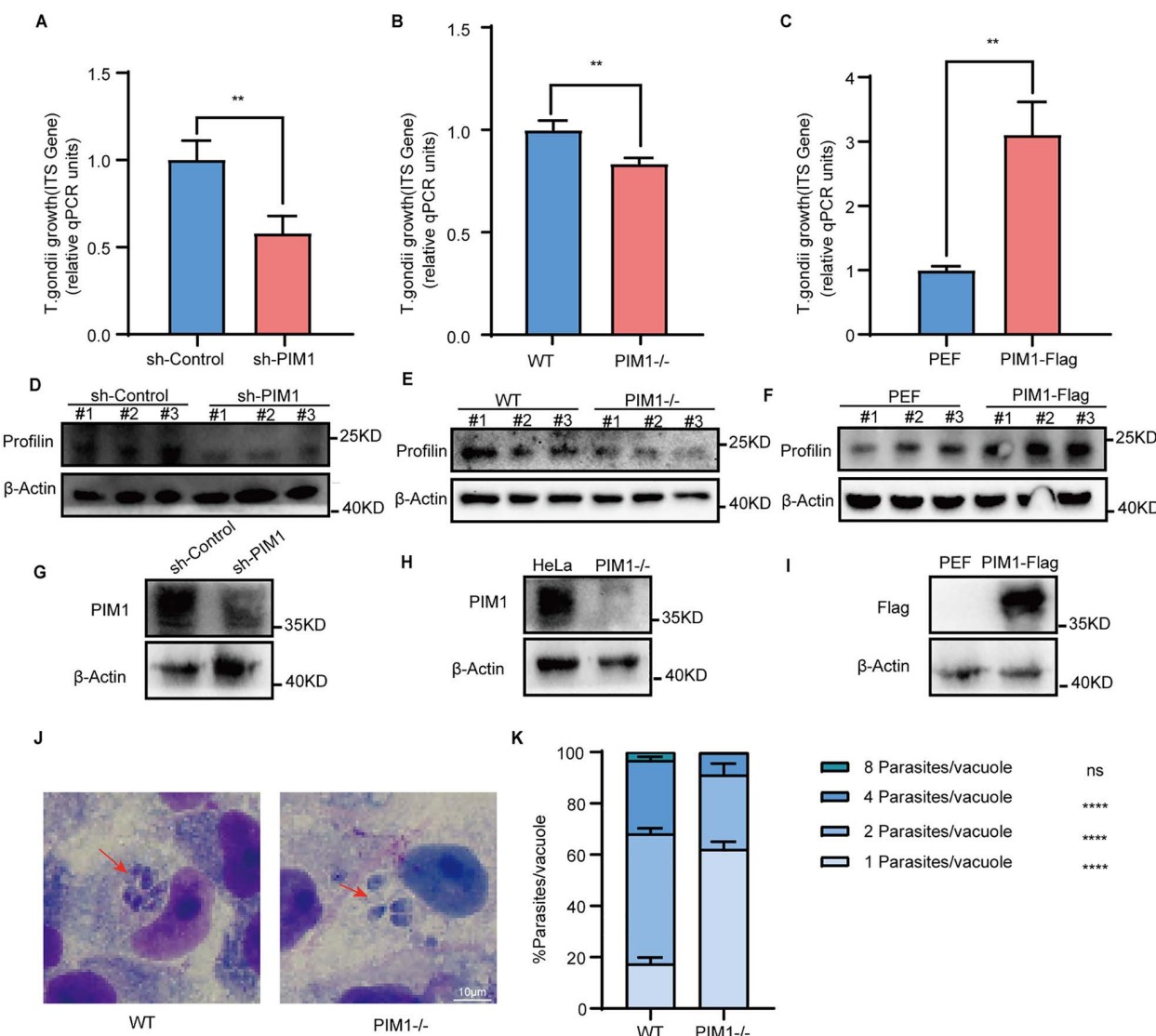

**Fig 1. PIM1 enhances the intracellular burden of *T. gondii* in infected cells.** (A–C) qRT-PCR quantification of TgITS-1 mRNA (normalized to GAPDH) in *T. gondii* ME49-infected cells (24 h p.i., MOI = 3). **(A)** RAW264.7 with stably transduced shRNA against PIM1 (sh-PIM1) vs. scrambled shRNA (sh-Control). **(B)** HeLa PIM1-knockout (PIM1-/-) vs. wild-type (WT) HeLa. **(C)** HEK293T transiently over-expressing Flag-tagged PIM1 (PIM1-Flag) vs. empty vector (PEF). **(D–F)** WB showing parasite profilin (upper band) and loading control β-actin (lower band). Each blot contains three lanes (# 1, # 2, # 3) that correspond to three independent biological replicates. **(D)** RAW264.7 sh-Control vs. sh-PIM1. **(E)** HeLa WT vs. PIM1-/-. **(F)** HEK293T PEF vs. PIM1-Flag. **(G–I)** WB confirming PIM1 expression levels in the corresponding cell lines described above. **(J)** Representative Giemsa-stained images (100 × oil immersion) of WT and PIM1-/- HeLa cells at 24 h p.i.; red arrows indicate PVs. **(K)** Quantification of tachyzoites per PV (≥100 PVs per condition, three independent experiments). Data are mean ± s.d.; significance by two-tailed unpaired Student's t-test (A-C) and two-way ANOVA with multiple comparison tests **(K)** (n = 3): ns, not significant, **P < 0.01, ****P < 0.0001.

level, indicating that PIM1 can also promote the proliferation of the classic Type I RH strain and the Chinese-specific TgCtWh6 strain.

Upon invasion of host cells, *T. gondii* encases itself within a specialized membrane structure known as the parasitophorous vacuole membrane (PVM). Utilizing Giemsa staining, we quantified the number of *T. gondii* organisms within PVs in both PIM1-null HeLa cells and their wild-type counterparts. Upon PIM1 knockout, the number of tachyzoites per PV

reduced remarkably compared to that in wild-type cells (Fig 1J). For ease of observation, we have conducted a quantitative analysis of the number of *T. gondii* within the PVs. (Fig 1K). These results showed that the intracellular proliferation of *T. gondii* is compromised in the absence of PIM1. These findings provide compelling evidence that both exogenous and endogenous PIM1 kinases significantly enhance the intracellular proliferation of *T. gondii*. These data indicate that the intracellular proliferation of *T. gondii* is compromised in the absence of PIM1, suggesting that PIM1 plays a pivotal role in facilitating the parasite's life cycle within the host cell.

**PIM1 promotes *T. gondii* proliferation in kinase activity-dependent manner**

To ascertain the necessity of PIM1's kinase activity for its role in facilitating *T. gondii* proliferation, we employed a kinase-dead mutant of PIM1. Lys67 (K67) lies within the catalytic pocket of PIM1 kinase and is essential for substrate binding and catalysis. Multiple studies have shown that mutation at this residue severely impairs kinase activity, abolishing the phosphorylation of downstream targets [49–51]. We therefore employed PIM1-K67R, a widely validated kinase-dead mutant, to investigate the role of PIM1 catalytic activity in *T. gondii* proliferation. Our experimental findings indicate that 293T cells transfected with this mutant were incapable of enhancing the levels of TgITS-1 mRNA, in contrast to cells overexpressing wild-type PIM1, which significantly amplified the expression of TgITS-1 mRNA when compared to those transfected with an empty vector (Fig 2A). Furthermore, the PIM1 kinase inhibitor AZD1208 was applied. Based on the instructions and relevant literature, we selected the middle value of its commonly used concentrations, 5 µM. It was found that when cells were treated with AZD1208 after ME49 infection, the levels of TgITS-1 mRNA were significantly reduced compared with the untreated group. (Fig 2B). Additionally, the profilin protein levels were examined, revealing that overexpression of PIM1 substantially increased the profilin protein levels, an effect not observed with the PIM1-K67R mutant (Fig 2C). Consistent with these observations, the inhibitory impact of AZD1208 on PIM1 kinase activity abrogated the promotion of proliferation (Fig 2D), thereby underscoring the indispensable nature of PIM1's kinase activity for its functional role in the context of *T. gondii* intracellular development. These results collectively highlight the pivotal role of PIM1's kinase activity in the intracellular life cycle of *T. gondii* and its potential as a therapeutic target for interventions against toxoplasmosis.

**PIM1 promotes the proliferation of *T. gondii* by inhibiting the level of apoptosis in the host cell**

In our preceding investigations, we discerned that PIM1 is instrumental in augmenting the proliferation of *T. gondii*. Subsequently, we endeavored to delineate the underlying mechanisms mediating its effects. The apoptosis of host cells infected by *T. gondii* is acknowledged to be a pivotal factor in bolstering the humoral immune response against the parasites. Thus, we scrutinized the modulations in cellular apoptosis levels during *T. gondii* infection under conditions of PIM1 downregulation. Our experimental observations in Raw264.7 cells, in which PIM1 was stably knocked down (Fig 3C), revealed a significant downregulation of the mRNA levels of the anti-apoptotic gene *bcl-2* and a concomitant upregulation of the mRNA levels of the pro-apoptotic gene *bax*, in comparison to the control group (Fig 3A and 3B).

The Terminal deoxynucleotidyl transferase dUTP Nick End Labeling (TUNEL) staining is a highly sensitive technique designed to detect DNA double-strand breaks that are exposed during the apoptosis process by recognizing the 3'-termini. This method is capable of identifying even minute quantities of cellular apoptosis. Employing TUNEL staining on HeLa cells with PIM1 knockout and the control group, *T. gondii* infection or not, we assessed cellular apoptosis. Our findings indicated that in the uninfected group, the absence of PIM1 was associated with an enhanced incidence of apoptosis (Fig 3D). Subsequent fluorescence analyses revealed that PIM1 depletion alone increases the percentage of TUNEL-positive cells without altering the mean fluorescence intensity (Fig 3E). We therefore propose that PIM1 intrinsically suppresses apoptosis in uninfected cells, and that this inhibitory effect is further potentiated upon *T. gondii* infection. Consistently, following infection, reduced PIM1 expression was associated with a further increase in apoptosis (Fig 3D&3E). Taken together, the TUNEL results support the conclusion that PIM1 promotes *T. gondii* proliferation by inhibiting host cell apoptosis.

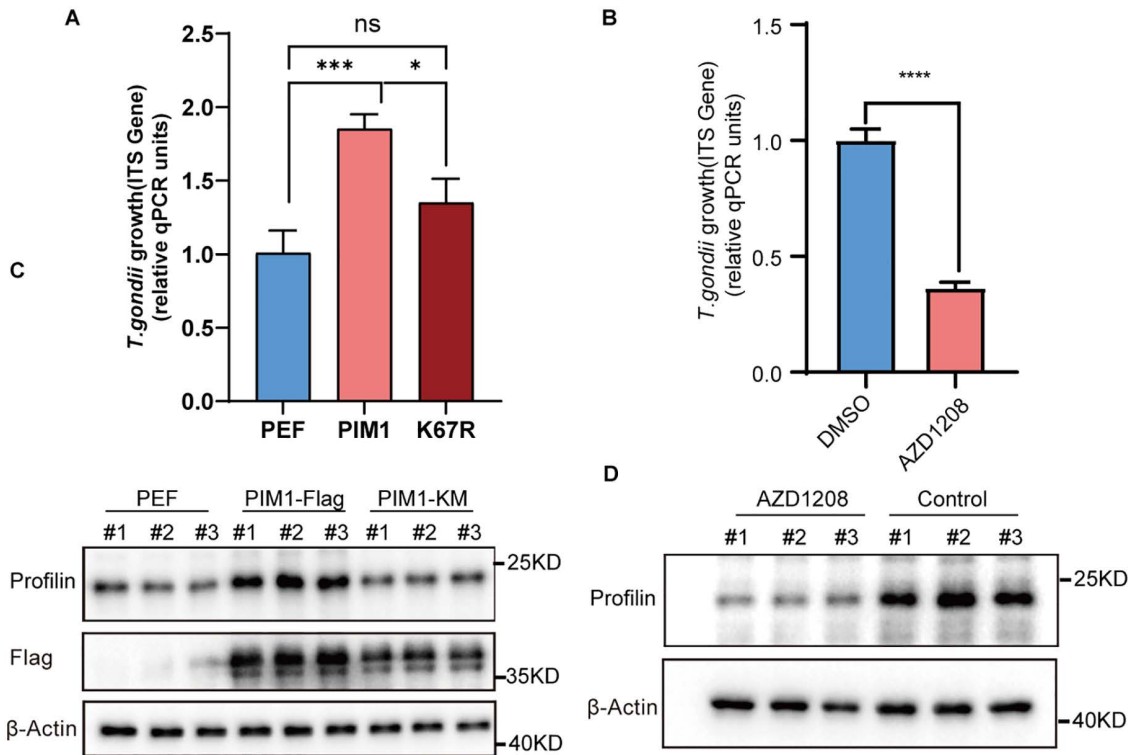

**Fig 2. PIM1 enhances the replication of *T. gondii* through a kinase activity-dependent mechanism.** (A) qRT-PCR quantification of TgITS-1 mRNA (normalized to GAPDH) in HEK293T cells transfected with empty vector (PEF), wild-type PIM1-Flag (WT), or kinase-dead PIM1-K67R-Flag (K67R), then infected with *T. gondii* ME49 (MOI = 3) and harvested 24 h post-infection. Values are mean ± s.d.; significance was determined by one-way ANOVA with multiple comparison tests: ns, not significant, *P < 0.05, ***P < 0.001. (B) qRT-PCR quantification of TgITS-1 mRNA (normalized to GAPDH) in RAW264.7 cells treated with DMSO (Control) or 5 μM AZD1208 for 2 h before infection, then infected with *T. gondii* ME49 (MOI = 3) and harvested 24 h post-infection. Values are mean ± s.d.; significance by two-tailed unpaired Student's t-test (n = 3): ****P < 0.0001. **(C, D)** WB for parasite profilin (upper band) and β-actin (loading control). Each set of three lanes (# 1, # 2, # 3) represents three independent biological replicates. **(C)** HEK293T: PEF, WT, or K67R; **(D)** RAW264.7: DMSO vs. AZD1208. Anti-Flag blot confirms PIM1 transfection efficiency.

Additionally, we quantified the protein levels of Bcl-2, Bax, Caspase3 and the apoptotic marker cleaved-Caspase3. It was observed that post-*T. gondii* infection, the downregulation of PIM1 corresponded with a diminution in Bcl-2 protein levels, concurrent with an elevation in Bax and cleaved-Caspase3 protein levels (Fig 3F). Subsequently, we preliminarily explored the mechanism by which PIM1 inhibits apoptosis within cells following ME49 infection. Through mitochondrial isolation, we found that upon infection, the suppression of PIM1 led to a reduction of cytochrome c within the mitochondria and an increase of cytochrome c released into the cytoplasm, suggesting that it may inhibit apoptosis through the mitochondrial pathway (Fig 3G). In aggregate, these data imply that PIM1 exerts an inhibitory influence on intrinsic apoptosis, potentially playing a critical role in promoting the intracellular survival and proliferation of *T. gondii*.

### AZD1208 inhibits the proliferation of *T. gondii in vivo*

In prior *in vitro* cellular assays, evidence has accrued to suggest that the serine/threonine kinase PIM1 is instrumental in the proliferation of *T. gondii*. The present study endeavors to elucidate whether this regulatory role extends to the *in vivo* context within murine models. Building upon the established mechanistic insights into PIM1's facilitation of *T. gondii* replication, we are poised to ascertain the congruence of this function in a physiological milieu. Leveraging the documented safety and efficacy profile of AZD1208 as an anti-neoplastic agent in mice, this investigation extends its application to the

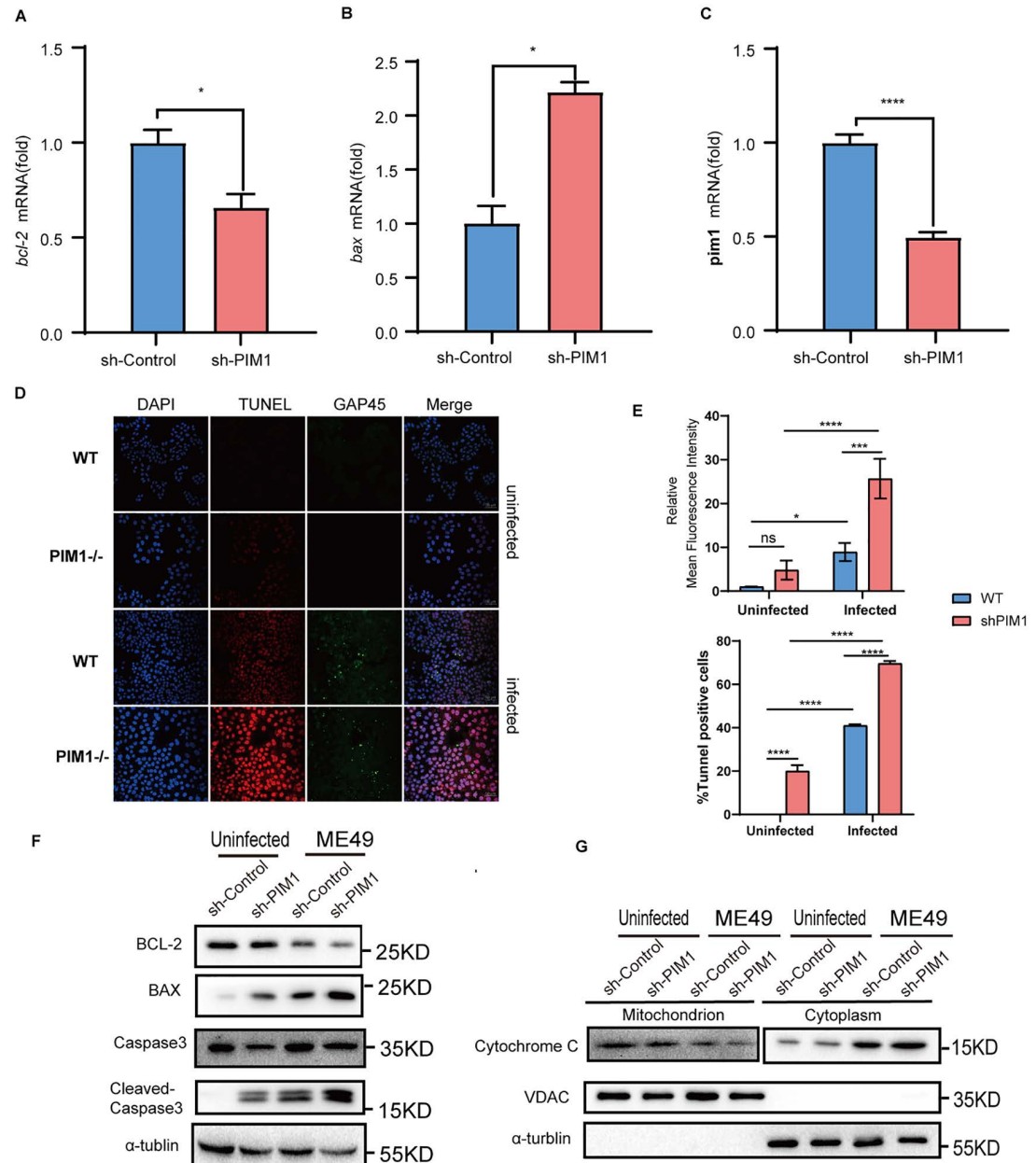

**Fig 3. PIM1 facilitates the induction of apoptosis in cells infected with *T. gondii*.** (A-C) qRT-PCR analysis was conducted to quantify the mRNA expression levels of *bcl-2*, *bax*, and *pim1* (normalized to GAPDH) in control (sh-Control) and PIM1 stably knocked down (sh-PIM1) RAW264.7 cells following *T. gondii* infection (24 h p.i., MOI = 3). **(D)** Representative images of TUNEL staining (red), DAPI (blue), and GAP45 of *T. gondii* (green) in uninfected or ME49-infected HeLa PIM1-knockout (PIM1-/-) and wild-type (WT) HeLa. **(E)** Quantitative analysis of mean fluorescence intensity and percentage of TUNEL-positive cells. **(F)** WB analysis of apoptosis-related proteins BCL-2, BAX, Caspase3 and cleaved Caspase3 in uninfected and ME49-infected cells with sh-Control or sh-PIM1 in Raw264.7 cells. α-tubulin was used as a loading control. **(G)** WB analysis of mitochondrial and cytosolic fractionation in uninfected and ME49-infected cells with sh-Control or sh-PIM1 in Raw264.7 cells. The mitochondrial proteins cytochrome c and the mitochondrial marker VDAC, as well as the cytosolic marker α-tubulin, were detected respectively. Values are mean ± s.d.; significance by two-tailed unpaired Student's t-test (A to C) and two-way ANOVA with multiple comparison tests **(E)** (n = 3): ns, not significant, *P < 0.05, **P < 0.01, ****P < 0.001, ****P < 0.0001.

realm of infectious disease research. Considering that it exerts its effects via kinase activity, we have maintained its therapeutic concentration consistent with that used in cancer treatment [46,52].

The experimental paradigm commenced with the inoculation of mice with ME49 tachyzoites of *T. gondii ME49* strain or not infected, followed by a regimen of daily intraperitoneal injections of AZD1208 or Pyrimethamine for a period of seven days. Splenic and hepatic tissues were excised for subsequent analysis of *T. gondii* burden and quantification of cytokine profiles indicative of anti-parasitic immune responses (Fig 4A). The data revealed a marked diminution in the tissue load of *T. gondii* within the spleen and liver of mice subjected to PIM1 inhibition, in stark contrast to the untreated control cohort (Fig 4B and 4G). Concurrently, we found that the positive control drug Pyrimethamine exhibited better antiparasitic effects (Fig 4B). Furthermore, the interrogation of cytokine expression post-inhibitor intervention disclosed a pronounced upregulation in the hepatic and splenic mRNA levels of IFN-γ (Fig 4C and 4H) and IL-12 (Fig 4D and 4I), cytokines pivotal to the immunological cascade against *T. gondii*. In contrast to the AZD1208 group, we found that the mRNA level of IFN-γ was not upregulated, while the level of IL-12 was significantly increased in Pyrimethamine group. We speculate that this difference may be due to the distinct mechanisms through which Pyrimethamine and AZD1208 exert their effects. Pyrimethamine is a classic antifolate drug that inhibits dihydrofolate reductase, thereby blocking the synthesis of nucleotides in parasites and directly acting on the parasites to inhibit their growth and replication. In contrast, AZD1208 is a PIM1 inhibitor that achieves its antiparasitic effect by promoting apoptosis in infected cells, primarily modulating the host's immune response rather than directly targeting the parasite. From the results, we also observed that Pyrimethamine exhibited stronger antiparasitic activity, while AZD1208 also showed a significant antiparasitic effect, leading us to believe that PIM1 still holds potential as a drug target against *T. gondii*, considering its role in cancer, the authors believe that, especially for some cancer patients, PIM1 inhibitors are a better option.

### AZD1208 induced apoptosis *in vivo*

Subsequent to the delineation of PIM1's role in the murine model, the study proceeded to dissect the underpinning mechanisms by which PIM1 may be orchestrating *T. gondii* proliferation. Drawing from cellular studies, PIM1 has been implicated in facilitating *T. gondii* replication by suppressing apoptosis, we hypothesized a conserved mechanism *in vivo*. To this end, RNA and protein extracts from the splenic tissues of the aforementioned murine model were subjected to qRT-PCR and WB analysis. Experimental data revealed that in ME49-infected mice treated with AZD1208, *bcl-2* mRNA was significantly downregulated (Fig 5B), whereas *bax* mRNA was upregulated compared to the control groups (Fig 5D). Similarly, in uninfected mice, AZD1208 treatment led to a decrease in *bcl-2* mRNA (Fig 5A) and an increase in *bax* mRNA relative to the control groups (Fig 5C). In the Pyrimethamine positive control group, we found that the mRNA level of *bcl-2* was decreased compared with the control, while the mRNA level of *bax* did not show significant changes. WB analysis showed that following ME49 infection, AZD1208-treated mice exhibited significantly reduced BCL-2 protein levels and significantly elevated BAX and cleaved-Caspase3 protein levels compared to the control and Pyrimethamine groups. In uninfected mice, BCL-2 protein levels were also lower, and BAX protein levels were higher in the AZD1208-treated group than in the control and Pyrimethamine groups, although the changes were less pronounced than in the infected group (Fig 5E). Collectively, the results suggest that AZD1208 treatment following ME49 infection correlates with increased apoptotic markers in murine spleen tissue, implying that PIM1 facilitates ME49 replication by suppressing apoptosis *in vivo*. However, the observed effects of PIM1 inhibition on normal cells highlight potential limitations of targeting PIM1 as an anti-parasitic strategy.

### Discussion

The development of therapeutic strategies against *T. gondii* has been challenging due to its intracellular lifestyle and its ability to evade the host immune response. The complexity of *T. gondii*'s interaction with its host is further highlighted by the parasite's ability to manipulate host cell machinery to its advantage. However, our identification of host kinase PIM1 as an essential driver of *T. gondii* proliferation provides a clear and druggable target.

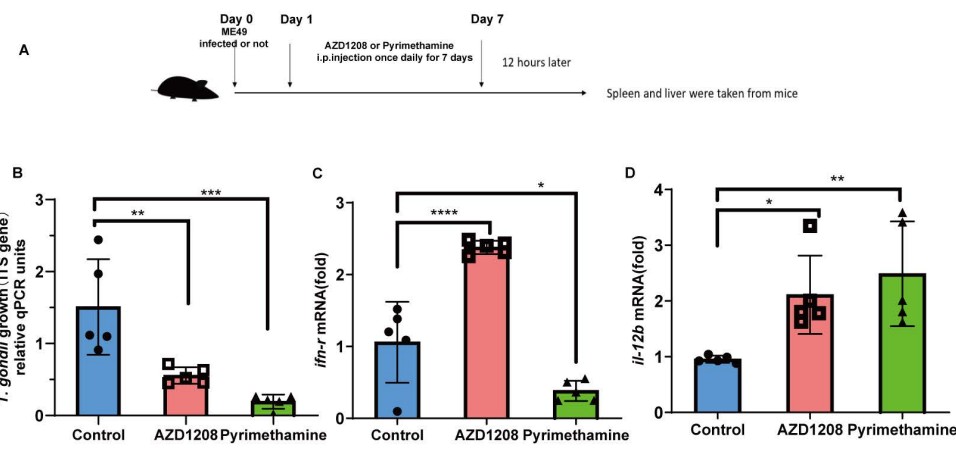

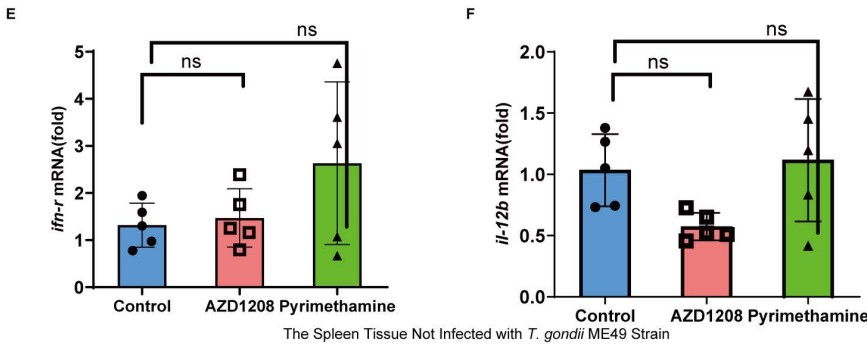

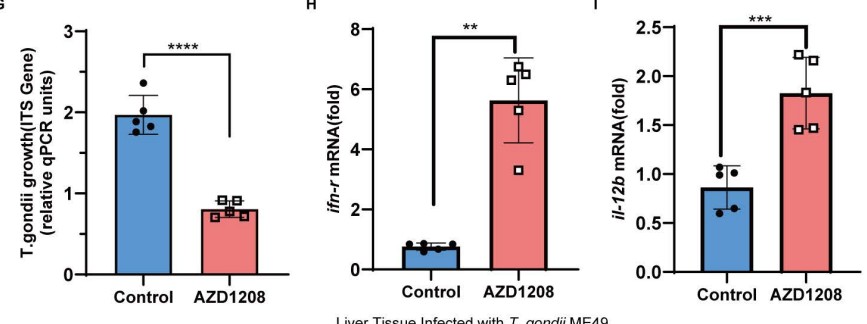

**Fig 4. PIM1 Inhibitor Diminishes Parasite Burden in Mice Infected with *T. gondii* and modulates cytokine expression in infected mice. (A)** Schematic of the experimental timeline for *T. gondii* ME49 strain infection and treatment with AZD1208 or Pyrimethamine. (B-D) qRT-PCR analysis was conducted to quantify the mRNA expression levels of TgITS-1*, Ifn-γ*, and *Il-12b* (normalized to GAPDH) in spleen tissues from ME49-infected mice treated with AZD1208 or Pyrimethamine compared to controls. (E-F) qRT-PCR analysis of *Ifn-γ* and *Il-12b* mRNA levels (normalized to GAPDH) in spleens from uninfected mice treated with AZD1208 or Pyrimethamine. (G-I) qRT-PCR analysis of Tg*ITS-1, Ifn-γ,* and *Il-12b* mRNA levels (normalized to GAPDH) in livers from ME49-infected mice treated with AZD1208 compared to controls. Values are mean ± s.d.; significance by one-way ANOVA with multiple comparison tests (B to F) and two-tailed unpaired Student's t-test (G to **I)** (n = 5): ns, not significant, *P < 0.05, **P < 0.01, ***P < 0.001, ****P < 0.0001.

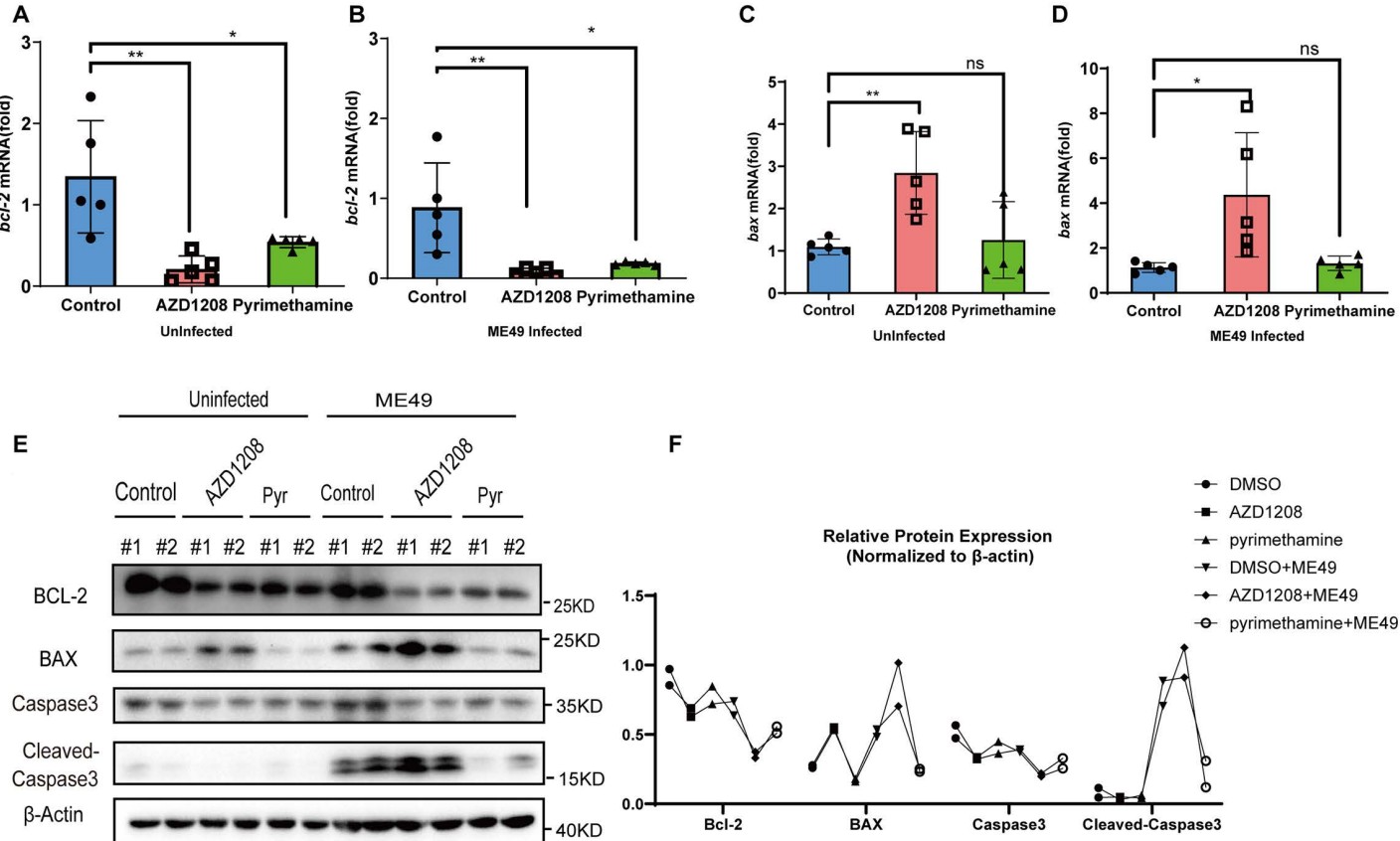

**Fig 5. PIM1 Inhibitor Promotes Apoptosis in Mice Infected with _T. gondii_.** (A-D) qRT-PCR was employed to assess the mRNA expression levels of _bcl-2_ and _bax_ in spleen from mice treated with DMSO (Control), AZD1208 and Pyrimethamine following _T. gondii_ infection or not. **(A)** Expression levels of bcl-2 mRNA in the infected group.(B) Expression levels of bcl-2 mRNA in the non-infected group. **(C)** Expression levels of bax mRNA in the infected group. **(D)** Expression levels of bax mRNA in the non-infected group. Values are mean ± s.d.; significance by one-way ANOVA with multiple comparison tests (n = 5): ns, not significant, *$P < 0.05$, **$P < 0.01$. **(E)** WB analysis was performed to assess the protein levels of apoptosis markers BCL-2, BAX, Caspase3, and cleaved-Caspase3 in splenic tissues from both uninfected and ME49-infected mice treated with DMSO, AZD1208, or Pyrimethamine. β-actin was used as a loading control. Each set of two lanes (# 1, # 2) represents two independent biological replicates. **(F)** A quantitative analysis of the WB band intensities in **C**.

Our findings underscore the significance of PIM1 in the intracellular survival and proliferation of the parasite. In vitro and _in vivo_ data establish it as a therapeutic target for toxoplasmosis. The inhibition of PIM1 activity, achieved through the use of the kinase inhibitor AZD1208, resulted in a significant reduction of _T. gondii_ load in murine models, highlighting the potential of PIM1 as a novel treatment strategy. It should be noted that the current study is primarily based on a single strain of _T. gondii_ (ME49). Extension to the type I RH strain (1.76 ± 0.56-fold, n = 3, p = 0.037) and the Chinese 1 Wh6 strain (1.92 ± 0.87, n = 3, p = 0.0128) revealed comparable proliferation-enhancing effects (S1 Fig; quantified by ImageJ), indicating functional conservation of PIM1 across genotypes. In addition, our study primarily focuses on the acute phase of infection. Whether PIM1 can still serve as an anti-parasitic target during chronic infection remains to be explored. Nevertheless, PIM1 remains a highly promising target for antiparasitic drugs. Mechanistically, subcellular fractionation showed that PIM1 knock-down significantly increases cytochrome-c release from mitochondria into the cytosol (Fig 3G), indicating that PIM1 suppresses host mitochondria-mediated apoptosis to create an intracellular niche permissive for parasite survival. Taken together, these cross-genotype data provide a pre-clinical rationale for advancing PIM1-targeted interventions

toward *in-vivo* validation and, ultimately, clinical evaluation. Given PIM1's role in cancer and its application as a target for anticancer drug screening, we have reason to believe that PIM1 inhibitors, as part of a combination therapy regimen, hold considerable potential, especially for immunocompromised individuals. This approach not only aids in cancer treatment but also concurrently eliminates *T. gondii* infection. Importantly, PIM1 inhibitors (e.g., AZD1208) induce apoptosis in both infected and uninfected host cells. Thus, targeting PIM1 in combination therapy is particularly rational for treating toxoplasmosis in cancer patients.

Frickel et al. showed that the kinase PIM1 phosphorylates GBP1 at Ser156, recruiting 14-3-3σ proteins that sequester GBP1 in the cytosol and thereby prevent host-membrane damage. During *T. gondii* infection the parasite effector TgIST blocks IFN-γ signaling, rapidly depleting PIM1 and unleashing GBP1 onto the parasitophorous vacuole [53]. Our findings extend rather than contradict this paradigm. We reveal that PIM1 also exerts a pro-parasitic function within infected cells: suppression of host apoptosis. Pharmacological or genetic ablation of PIM1 markedly reduces parasite burden while increasing host-cell death, indicating that *T. gondii* maintains PIM1 activity to secure an anti-apoptotic niche. Thus, PIM1 operates bidirectionally during infection: in uninfected bystander cells it phosphorylates GBP1 to limit membrane damage, and in infected cells PIM1 performs a dual role—on the one hand, the parasite effector TgIST blocks IFN-γ signaling, rapidly depleting PIM1 and unleashing GBP1 onto the parasitophorous vacuole; on the other hand, it inhibits apoptosis, creating a survival-permissive environment for the parasite. The Toxoplasma protein(s) that interact with PIM1 to modulate apoptosis are currently under investigation and will illuminate how the parasite hijacks host signaling for its own benefit.

In conclusion, our research highlights the importance of PIM1 in the complex host-parasite relationship with *T. gondii*. The inhibition of PIM1 activity presents a promising avenue for the development of targeted therapies against toxoplasmosis, with potential implications for the treatment of other diseases influenced by this protozoan parasite (S2 Fig).

## Supporting information

**S1 Fig. Effect of PIM1 depletion on proliferation of *T. gondii* RH strain and Chinese 1 genotype Wh6 strain.** Depletion of PIM1 dampens the proliferation of the RH strain and the Chinese 1 genotype Wh6 strain of *T. gondii*. In HeLa WT and PIM1-/- cells, WB showing parasite profilin (upper band) and loading control β-actin (lower band) after *T. gondii* infection at MOI = 3 for 24 hours. Each blot contains three lanes (#1, #2, #3) corresponding to three independent biological replicates.
(TIF)

**S2 Fig. Graphical abstract summarizing the main findings of this study.** The character of PIM1 in regulating proliferation of *T. gondii*. PIM1 facilitated the proliferation of *T. gondii* via suppressing apoptosis. AZD1208, a small molecule inhibitor of PIM1, led to the elimination of *T. gondii* and consequently reduced the parasite load.
(TIF)

**S1 Table. Primer sequences used for qPCR in this study.**
(DOCX)

**S1 Data. Excel file containing the raw numerical data used to generate all figures.**
(XLSX)

**S2 Data. Original Western blot images.**
(RAR)

**S3 Data. Western blot images with molecular weight markers indicated.**
(RAR)

## Acknowledgments

The anti-*T. gondii* antibodies including Profilin and GAP45 were generously provided by Prof. Yonggen Jia.

## Author contributions

**Conceptualization:** Lijie Pan, Li Yu, Yuanyuan Cao.

**Data curation:** Fan Zhang, Yun Yang.

**Funding acquisition:** Wen Zhang, Yuanyuan Cao.

**Investigation:** Lijie Pan, Fan Zhang, Yun Yang, Yue Sun, Lingling Song.

**Methodology:** Lijie Pan, Fan Zhang, Famin Zhang, Jinjin Zhu, Yang Wang, Qingli Luo.

**Resources:** Qingli Luo.

**Software:** Chong Wang, Wen Zhang.

**Supervision:** Wen Zhang, Li Yu, Yuanyuan Cao.

**Writing – original draft:** Yue Sun, Wen Zhang, Yuanyuan Cao.

**Writing – review & editing:** Li Yu, Yuanyuan Cao.

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
