## [Decision Letter · Decision Letter 0]

28 Apr 2025

Unraveling the role of host kinase PIM1 in *Toxoplasma gondii* infection: implications for therapies

Dear Dr. Cao,

Thank you for submitting your manuscript to PLOS Neglected Tropical Diseases. After careful consideration, we feel that it has merit but does not fully meet PLOS Neglected Tropical Diseases's publication criteria as it currently stands. Therefore, we invite you to submit a revised version of the manuscript that addresses the points raised during the review process.

Please submit your revised manuscript within 60 days Jun 27 2025 11:59PM. If you will need more time than this to complete your revisions, please reply to this message or contact the journal office at plosntds@plos.org. Please include the following items when submitting your revised manuscript:

We look forward to receiving your revised manuscript.

Kind regards,

Laura-Isobel McCall

Section Editor

Laura-Isobel McCall

Section Editor

Shaden Kamhawi

co-Editor-in-Chief

Paul Brindley

co-Editor-in-Chief

**Additional Editor Comments:**

While the reviewers were interested and appreciated the value of this work, they noted several major issues with the quality of the qPCR, Western blot and Tunnel assay data that must be addressed. While experiments with additional parasite strains would be desirable, alternatively these comments can be addressed by changes in the discussion. All other reviewer comments should be addressed.

**Journal Requirements:**

3) We note that your Data Availability Statement is currently as follows: "All relevant data are within the manuscript and its Supporting Information files". Please confirm at this time whether or not your submission contains all raw data required to replicate the results of your study. Authors must share the “minimal data set” for their submission. PLOS defines the minimal data set to consist of the data required to replicate all study findings reported in the article, as well as related metadata and methods (https://journals.plos.org/plosone/s/data-availability#loc-minimal-data-set-definition).

- The points extracted from images for analysis..

4) Please amend your detailed Financial Disclosure statement. This is published with the article. It must therefore be completed in full sentences and contain the exact wording you wish to be published.

**Reviewers' Comments:**

Reviewer's Responses to Questions

**Key Review Criteria Required for Acceptance?**

**Methods**

-Are the objectives of the study clearly articulated with a clear testable hypothesis stated?

-Is the study design appropriate to address the stated objectives?

-Is the population clearly described and appropriate for the hypothesis being tested?

-Is the sample size sufficient to ensure adequate power to address the hypothesis being tested?

-Were correct statistical analysis used to support conclusions?

-Are there concerns about ethical or regulatory requirements being met?

Reviewer #1: See below comments

Reviewer #2: While the study objectives are generally well-defined, it would be beneficial to explicitly state the hypothesis in terms of specific molecular mechanisms by which PIM1 influences T. gondii proliferation. The study design, combining in vitro and in vivo approaches, is appropriate, but the lack of clarity regarding the duration of mouse infection compromises reproducibility. The cell populations and animal model are described, but justifying the choice of the ME49 strain and discussing the generalization of the results to other strains would be important. The sample size (n=3) suggests adequate power, but a brief justification based on statistical power analysis would be helpful. The statistical tests (t-test) seem appropriate, but more details and justifications are needed. Finally, a more detailed statement regarding compliance with ethical guidelines for animal research is essential.

Reviewer #3: The objectives of the study is articulated well and is timely with a clear hypothesis. However, the study design could be more robust, for instances, authors should use type I and type III strains of Toxoplasma to show that the effect of PIM1 is not specific to host response against a specific Toxoplasma strain. Also, authors should also test the TgIST1 KO parasites as it can activate the host IFN-responses much robustly compared to the Wild-type parasites.

Statistical tests were done correctly and ethical requirements were met.

**Results**

-Does the analysis presented match the analysis plan?

-Are the results clearly and completely presented?

-Are the figures (Tables, Images) of sufficient quality for clarity?

Reviewer #1: See below comments

Reviewer #2: In general, the analysis presented corresponds to the plan outlined in the methodology. The authors use qRT-PCR, Western blot, Giemsa staining, and TUNEL staining to assess T. gondii proliferation and cellular apoptosis, as described. However, the lack of details about the statistical tests used makes it difficult to fully assess the appropriateness of the analysis.

The results are presented generally clearly, with figures and tables that illustrate the main findings. However, some additional information would be helpful to increase clarity:

Indicate the internal control genes used in the qRT-PCR graphs.

Mention the concentration of the AZD1208 inhibitor used and the justification for this choice.

Provide information about the number of cells analyzed in each experiment (e.g., Giemsa staining, TUNEL staining)

The figures are of good quality and easy to interpret.

Suggestions:In the qRT-PCR results (Figures 1A, 1B, 1C), indicate which genes were used as internal controls (e.g., GAPDH).

In Figure 1J, increase the resolution of the images for better visualization of the parasites.

Reviewer #3: The analysis and the interpretation of the data needs a lot of improvement. Firstly, authors need a consistent data throughout. For example, authors wrote in HeLa cells, they use Sh-PIM1 in Fig. 1A but then in Fig. 1J they used PIM1 KO HeLa cells! which one is true KO or knockdown using Sh RNA?

Secondly, Western blot pictures for Cleaved Caspase 3 is not convincing at all and same for all other markers. Authors should show uninfected, WT and uninfected knockdown cells for all the markers as well as the total Caspase 3.

For all the western blot picture, full unprocessed blots should be given in supplementary materials.

Thirdly, Tunnel assay pictures are not at all convincing. Authors should show with higher magnification and also to show that the parasites are infected in the cells using SAG1/GRA7 staining in parallel for which commercially available antibodies are there.

In Figure 5, the loading of different wells are extremely unequal and is of poor quality.

Finally, throughout most of the experiments, critical controls are missing. How do we know the effect of the drugs in the uninfected mice also!! and also, authors should use a positive control drug which will negate the effects.

**Conclusions**

-Are the conclusions supported by the data presented?

-Are the limitations of analysis clearly described?

-Do the authors discuss how these data can be helpful to advance our understanding of the topic under study?

-Is public health relevance addressed?

Reviewer #1: See below comments

Reviewer #2: The authors mention some limitations, such as the need to identify the T. gondii protein that interacts with PIM1. However, other important limitations are not addressed:

The generalization of the results to other strains of T. gondii.

The lack of investigation of the detailed molecular mechanisms by which PIM1 regulates apoptosis.

The absence of data on the efficacy of the AZD1208 inhibitor in chronic infection models.

The conclusions about the role of PIM1 in T. gondii proliferation and modulation of apoptosis are generally supported by the data presented, both in vitro and in vivo. However, it is important to note that the conclusions are primarily based on a single strain of T. gondii (ME49), it should be mentioned.

The authors discuss how the identification of PIM1 as a therapeutic target may open new avenues for the development of treatments against toxoplasmosis. They also mention the relevance of PIM1 in modulating the host immune response. However, they could further explore the translational potential of these findings, considering the development of combination therapies or the use of PIM1 inhibitors in conjunction with other drugs.

The authors mention the high prevalence of toxoplasmosis and the impact on immunocompromised individuals. However, they could further emphasize the relevance to public health by discussing:

The economic impact of toxoplasmosis (treatment costs, loss of productivity).

The importance of preventing toxoplasmosis in pregnant women.

The need to develop new therapies to overcome resistance to current treatments.

The conclusions are generally supported by the data, but generalization is limited by the use of a single T. gondii strain. The authors mention some limitations, but do not address the generalization of the results, the detailed molecular mechanisms, and the efficacy in chronic infection models. The discussion about advancing understanding of the topic is good, but the translational potential could be further explored. The relevance to public health is mentioned, but the economic impact, prevention in pregnant women, and the need for new therapies could be emphasized.

Reviewer #3: Although the data is interesting, but currently many data is questionable which needs clear clarification and controls

**Editorial and Data Presentation Modifications?**

Reviewer #1: (No Response)

Reviewer #2: Yes, I recommend the following editorial and data presentation modifications:

Revise the writing of the introduction to make it more concise and focused, avoiding excessively generic information about T. gondii.

Indicate which genes were used as internal controls in the qRT-PCR results (Figures 1A, 1B, 1C).

Mention the concentration used of the AZD1208 inhibitor and the justification for this choice in the results.

Improve the resolution of the microscopy images (e.g., Figure 1J) for better visualization of the parasites."

Recommendation:

Given the other shortcomings we identified (lack of clarity about the duration of mouse infection, need for more support in the discussion), "Major Revisions" is the most appropriate recommendation.

Reviewer #3: NA

**Summary and General Comments**

Reviewer #1: This manuscript by Pan L and colleagues reports host protein, Moloney murine leukemia virus 1 protein kinase (PIM1), its role in pathogenesis of Toxoplasma gondii infection and its potential as therapeutic target for treatment of T. gondii infections. They found that PIM1 inhibited apoptosis of T. gondii-infected host cells, which lead to enhancing intracellular proliferation of T. gondii. They further showed in a mouse model that the PIM1 inhibitor AZD1208 treated mice harbored lighter parasitic load upon T. gondii infection than the non-treated controls. They enthusiastically concluded that PIM1 is a promising target for the development of anti-T. gondii therapeutics.

The most important concern of this reviewer is that targeting the host protein PIM1 as therapeutics. For cancer chemotherapies, this may be considered, but not for T. gondii infections. PIM1 inhibitors such as AZD1208 very likely cause apoptosis of not only T. gondii infected host cells but also normal non-infected host cells. The line of in vitro evidence strongly suggests that AZD1208-treated mice of non-T. gondii infection would suffer from consequence of apoptosis, counteracting its usefulness as therapeutic agents. The authors must redirect their attention to the roles of PIM1 on the molecular pathogenesis of T. gondii.

In mouse model experiments, one important control is missing, which is non-T. gondii infected animals with AZD1208 treatment. This would show whether such a treatment is specifically to only T. gondii-infected host cells. This is pertinent even PIM1 is remotely considered as a therapeutic target.

An additional aspect the authors need to pay attention to is figure legends. Figure legends should be able to stand alone by themselves for readers to understand the figures without reference to main text. These include brief descriptions of methods, clear labels and what each label is, which are lack in the legends of this manuscript. What are the lanes of Western blots of the figures? There are multiple lanes with no explanation what they are.

Some methods are not presented, such as quantification of tachyzoites, protein extraction. Please see the PDF file for more information.

Reviewer #2: The manuscript addresses a relevant topic within the scope of the journal, investigating the role of host kinase PIM1 in Toxoplasma gondii infection. Identifying host factors that influence parasite proliferation can open new avenues for developing therapies. Overall, the study is well-conducted and presents interesting results. However, some areas need revision to increase the clarity and impact of the work.

Strengths:

Relevance: The study focuses on a neglected tropical disease of great global importance.

Originality: The investigation of the role of PIM1 in T. gondii infection is an original contribution.

Methodology: The study employs a combination of in vitro and in vivo approaches.

Results: The results are clear and well-presented, with robust quantitative data.

Weaknesses:

Introduction: Could be more concise and focused, avoiding excessively generic information about T. gondii.

Move the information about the different strains of T. gondii (lines 40-47) to a later paragraph, where the relevance to the study can be better established.

In line 53, specify what the "critical need" for new therapies is (resistance, side effects, etc.).

The last sentence of the introduction (lines 95-97) is a bit vague. Better specify which "potential targets" were identified.

The introduction (lines 40-47) presents a simplified view of the genetic diversity of T. gondii by mentioning only the three classic lineages (Type I, II, and III). Although this classification has been historically relevant, it does not reflect the complexity of the parasite's population structure, which includes a variety of other genotypes and lineages with different virulence characteristics and geographic distribution. It is recommended to expand this section to include a more comprehensive discussion about the genetic diversity of T. gondii, citing recent studies that have investigated the parasite's population structure in different regions of the world (e.g., Ajzenberg et al., 2004; Lehmann et al., 2006; Su et al., 2012). In addition, it is suggested to mention the importance of genetic diversity in the parasite's adaptation to different hosts and environments, as well as in the clinical manifestations of toxoplasmosis.

Discussion: Some statements need more support in the existing literature.

Writing: There is room for improvement in the fluency and clarity of the text.

the following additional analyses and clarifications are required:

Hypothesis: Explicitly state the study's hypothesis in terms of specific molecular mechanisms by which PIM1 influences T. gondii proliferation (e.g., detailing the impact on apoptosis pathways).

Study Design: Correct the lack of clarity regarding the duration of mouse infection with T. gondii ME49 prior to the start of treatment with DMSO or AZD1208. This information is crucial for reproducibility. Indicate which genes were used as internal controls in the qRT-PCR results (Figures 1A, 1B, 1C).

Mention the concentration used of the AZD1208 inhibitor and the justification for this choice in the results.

Improve the resolution of the microscopy images (e.g., Figure 1J) for better visualization of the parasites."

Recommendation:

Given the other shortcomings we identified (lack of clarity about the duration of mouse infection, need for more support in the discussion),

Population: Justify the choice of the ME49 strain (Type II) of T. gondii as the model for this study. Discuss whether the results obtained with this strain can be generalized to other strains of T. gondii, considering potential differences in virulence and host interactions.

Sample Size: Include a brief justification for the sample size (n=3) used in each experiment. Ideally, this should include a reference to a statistical power analysis performed to ensure adequate power to detect meaningful differences.

Statistical Analysis: Provide more specific details regarding the statistical tests used. Specify whether t-tests were paired or unpaired, and whether assumptions of normality were tested. Justify the choice of statistical tests based on the characteristics of the data.

Ethics: Include a more detailed statement regarding compliance with relevant ethical and regulatory guidelines for animal research. This should include the specific guidelines followed and any measures taken to minimize animal suffering.

Indicate which genes were used as internal controls in the qRT-PCR results (Figures 1A, 1B, 1C).

Mention the concentration used of the AZD1208 inhibitor and the justification for this choice in the results.

Improve the resolution of the microscopy images (e.g., Figure 1J) for better visualization of the parasites.

Reviewer #3: The manuscript is interesting but the data is not having good controls and inconclusive representation. Hence, I recommend major revision for the article.

PLOS authors have the option to publish the peer review history of their article (what does this mean? ). If published, this will include your full peer review and any attached files.

**Do you want your identity to be public for this peer review?** For information about this choice, including consent withdrawal, please see our Privacy Policy .

Reviewer #1: No

Reviewer #2: No

Reviewer #3: No

**Figure resubmission:**

**Reproducibility:**



---

## [Decision Letter · Decision Letter 1]

9 Oct 2025

Unraveling the role of host kinase PIM1 in *Toxoplasma gondii* infection: implications for therapies

Dear Dr. Cao,

Thank you for submitting your manuscript to PLOS Neglected Tropical Diseases. After careful consideration, we feel that it has merit but does not fully meet PLOS Neglected Tropical Diseases's publication criteria as it currently stands. Therefore, we invite you to submit a revised version of the manuscript that addresses the points raised during the review process.

Please submit your revised manuscript within 60 days Nov 08 2025 11:59PM. If you will need more time than this to complete your revisions, please reply to this message or contact the journal office at plosntds@plos.org. Please include the following items when submitting your revised manuscript:

We look forward to receiving your revised manuscript.

Kind regards,

Laura-Isobel McCall

Section Editor

Laura-Isobel McCall

Section Editor

Shaden Kamhawi

co-Editor-in-Chief

Paul Brindley

co-Editor-in-Chief

**Additional Editor Comments :**

Overall, reviewers appreciated your efforts in revising the paper. However, some issues remain from reviewers 2 and 3 that must be addressed. Reviewer 3 particularly noted concerns with regards to PIM1 expression levels being higher than expected. Authors should provide details on how protein identity in Western blots was confirmed, since many of the raw Western blot files provided lack molecular weight markers.

**Journal Requirements:**

1) Please ensure that the Title in your manuscript file is written correctly.

Note: Please correct the spelling of this word "implicatof ions" as the title in the manuscript is currently as follows: "Unraveling the role host kinase PIM1 in Toxoplasma gondii infection: implicatof ions for therapies."

2) Please amend your detailed Financial Disclosure statement. This is published with the article. It must therefore be completed in full sentences and contain the exact wording you wish to be published.

**Reviewers' Comments:**

Reviewer's Responses to Questions

**Key Review Criteria Required for Acceptance?**

**Methods**

-Are the objectives of the study clearly articulated with a clear testable hypothesis stated?

-Is the study design appropriate to address the stated objectives?

-Is the population clearly described and appropriate for the hypothesis being tested?

-Is the sample size sufficient to ensure adequate power to address the hypothesis being tested?

-Were correct statistical analysis used to support conclusions?

-Are there concerns about ethical or regulatory requirements being met?

Reviewer #1: (No Response)

Reviewer #2: Please see Summary and General Comments

Reviewer #3: Statistical analysis are not correct. Following should be done -

1. For Figure 1K, Two Way ANOVA with multiple comparison tests(as it has two variables: WT vs KO cells and different number of parasites/vacuoles)

2. Figure 2A, One Way ANOVA (Three groups) with multiple comparison tests

3. Figure 3E, Two Way ANOVA with multiple comparison tests similar reasons as stated for Figure 1K

4. Figures 4B-F: All of them should be done using One Way ANOVA with multiple comparison tests

5. Figures 5A-D: All of them should be done using One Way ANOVA with multiple comparison tests

**Results**

-Does the analysis presented match the analysis plan?

-Are the results clearly and completely presented?

-Are the figures (Tables, Images) of sufficient quality for clarity?

Reviewer #1: (No Response)

Reviewer #2: Please see Summary and General Comments

Reviewer #3: In Figure 3, it is clearly appears from the representative images (both from previous and the new version) that just by depleting PIM1, cells are vulnerable to die and authors should provide the graph for % Tunnel positive cells too rather than just showing MFI! and discuss the result accordingly. Also, Still its not clear about the mechanism by which PIM1 mediates the protection of the parasites, a clear discussion is necessary.

Another surprising fact is that, PIM1 is inducible protein (by IFN and other cytokines) and the baseline expression is low, not only that, its an extremely short lived protein. I am surprised that how authors have detected such a thick band in WT cells? without any proteasome inhibitors! Authors need to clarify.

Furthermore, one of the major problem is most of the western blot picture are not having any molecular weight marker, how one can understand that the band appears at correct molecular weight size? I made this comment in the earlier version too, but authors did not answer to that point.

**Conclusions**

-Are the conclusions supported by the data presented?

-Are the limitations of analysis clearly described?

-Do the authors discuss how these data can be helpful to advance our understanding of the topic under study?

-Is public health relevance addressed?

Reviewer #1: (No Response)

Reviewer #2: Please see Summary and General Comments

Reviewer #3: It is surprising to see that the authors did not cite the paper that discover the role of PIM1 in Toxoplasma infection. PIM1 controls GBP1 activity to limit self-damage and to guard against pathogen infection” (Science, 2023) by Frickel et al. The discussion should be robust and unbiased. Authors are encouraged to discuss (similarities/differences) with this study to make the discussion more comprehensive.

**Editorial and Data Presentation Modifications?**

Reviewer #1: (No Response)

Reviewer #2: Please see Summary and General Comments

Reviewer #3: (No Response)

**Summary and General Comments**

Reviewer #1: The authors have adequately addressed the concerns of this reviewer in this revision.

Reviewer #2: Dear authors,

The manuscript has improved considerably after review; it is now clearer, more instructive, and more likely to be read and cited. I would like to congratulate you on the care with which you addressed the previous suggestions. Nevertheless, I still have a few specific suggestions that will bring greater clarity and precision of information to the work, particularly for researchers working on toxoplasmosis, in both experimental and population-based studies.

I suggest:

1-To include in the abstract that findings are limited to the acute phase of infection, underscoring the need for further studies in chronic-phase models.

2-To change the Introduction (lines 59-61): The statement ‘given the increasing drug resistance’ may overstate the current evidence. Clinical resistance in T. gondii is still rare, with only isolated reports and in-vitro studies demonstrating selectable resistance. I recommend rephrasing to emphasize the toxicity of current drugs and the potential (rather than established increase) for resistance, unless additional references are provided to support the claim of increasing resistance.

3-To change the Introduction (lines 73-75): I recommend replacing ‘atypical’ with ‘non-archetypal’. The term ‘atypical’ reflects only the historical order in which lineages I–III were first described and can imply that other lineages are rare exceptions, which is not the case in the regions mentioned. ‘Non-archetypal’ is more neutral, widely used in recent literature, and better conveys that these isolates represent the true breadth of the species’ genetic diversity, now grouped into multiple haplogroups (e.g., HG4–HG16). This change improves both accuracy and clarity without altering the original meaning.

4-To improve conciseness by splitting the overly long sentence in lines 513-518, as well as the one in 536-539, and by deleting repeated phrases such as “parasite within its host cells” and the duplicate mention of “PIM1 inhibitors.”

5-For better cohesion, add a short transition that links the main results to their clinical implications.

6-In the GBP1 paragraph, state explicitly how this pathway connects to your central finding on PIM1.

7-Methodologically, since the Western-blot data for the RH and Wh6 strains appear only in the Supplementary Material, include a brief quantitative summary in the Discussion (e.g., proliferation increased A–B ×).

8-Finally, address language issues: change the unnecessary capital letter in “Our study” (line 531), insert a hyphen in “proliferation-enhancing effects” (line 538), and replace “Noteworthy,” with “Importantly,” or “Of note,” (line 544).

Reviewer #3: Stated above

PLOS authors have the option to publish the peer review history of their article (what does this mean? ). If published, this will include your full peer review and any attached files.

**Do you want your identity to be public for this peer review?** For information about this choice, including consent withdrawal, please see our Privacy Policy .

Reviewer #1: No

Reviewer #2: No

Reviewer #3: No

**Figure resubmission:**
---

## [Decision Letter · Decision Letter 2]

6 Jan 2026

Dear Lecturer Cao,

We are pleased to inform you that your manuscript 'Unraveling the role of host kinase PIM1 in *Toxoplasma gondii* infection: implications for therapies' has been provisionally accepted for publication in PLOS Neglected Tropical Diseases.

Best regards,

Susan Madison-Antenucci, PhD

Section Editor

Laura-Isobel McCall

Section Editor

Shaden Kamhawi

co-Editor-in-Chief

Paul Brindley

co-Editor-in-Chief

Reviewer's Responses to Questions

**Key Review Criteria Required for Acceptance?**

**Methods**

-Are the objectives of the study clearly articulated with a clear testable hypothesis stated?

-Is the study design appropriate to address the stated objectives?

-Is the population clearly described and appropriate for the hypothesis being tested?

-Is the sample size sufficient to ensure adequate power to address the hypothesis being tested?

-Were correct statistical analysis used to support conclusions?

-Are there concerns about ethical or regulatory requirements being met?

Reviewer #2: Are the objectives of the study clearly articulated with a clear testable hypothesis stated?

The overarching aim is clear; add an explicit testable hypothesis at the end of the Introduction (e.g., PIM1 promotes T. gondii proliferation by suppressing host apoptosis, and pharmacologic inhibition reduces parasite load in vivo).

Is the study design appropriate to address the stated objectives?

Yes. The combined in vitro gain-/loss-of-function, kinase-dead mutant, and in vivo inhibitor approach is appropriate for the stated objectives.

Is the population clearly described and appropriate for the hypothesis being tested?

Cell lines and mouse strain are specified; clarify the in vivo timeline (infection day, dosing start/frequency/duration, harvest) in one place and separate ethics from methods text, however, please make explicit that all data derive from the acute post infection phase (tachyzoite stage), add a clear infection/treatment timeline, and state that chronic bradyzoite tissue cysts were not assessed

Is the sample size sufficient to ensure adequate power to address the hypothesis being tested?

Authors should specify biological n per group for in vivo endpoints, distinguish biological vs technical replicates, and provide a brief rationale/power note or acknowledge as a limitation.

Were correct statistical analyses used to support conclusions?

Where unpaired two‑tailed t‑tests are used with small n, report normality checks and any multiple‑comparison control; consider non‑parametric tests when assumptions are uncertain.

Are there concerns about ethical or regulatory requirements being met?

No. Requirements appear met; address a minor formatting fix by presenting a standalone Ethics statement (committee, protocol ID, guidelines).

Reviewer #3: (No Response)

**Results**

-Does the analysis presented match the analysis plan?

-Are the results clearly and completely presented?

-Are the figures (Tables, Images) of sufficient quality for clarity?

Reviewer #2: The analysis presented match the analysis plan. The analyses align with the objective (kinase dependence and in vivo inhibition) and support the mechanistic claim.

The results are clearly and completely presented? Partially. Correct the Figure 3 caption to reflect that PIM1 inhibits (rather than induces) apoptosis, and ensure n and replicate type are indicated across panels.

Most figures (Tables, Images) are adequate; ensure high resolution, consistent scale bars/labels, and uniform terminology.

Please annotate figures/legends to indicate that results correspond to the acute phase and that chronic/cyst stages were not evaluated.

Reviewer #3: (No Response)

**Conclusions**

-Are the conclusions supported by the data presented?

-Are the limitations of analysis clearly described?

-Do the authors discuss how these data can be helpful to advance our understanding of the topic under study?

-Is public health relevance addressed?

Reviewer #2: The conclusions are supported by the data presented; the central claim is substantiated. Authors could please temper the wording to “reduces parasite load/decreases the proportion of infected cells” rather than “eliminates infected cells,” to match the assays.

Regarding the limitations of the analysis, please expand to note the use of a single lineage (type II/ME49), the short time frame and small n, potential off target and pharmacokinetic (PK) considerations for AZD1208, and the limited generalizability to chronic/cyst stages.

The authors discuss how these data advance understanding; this could be strengthened by briefly outlining translational implications (host directed therapy and combinations with standard of care).

Regarding public health relevance, please add a concise framing of disease burden, at risk populations, and how host directed PIM1 inhibition could complement current therapies.

Conclusions should be limited to acute infection; please temper language regarding therapeutic implications and avoid implying cure or effects on chronic tissue cysts

Reviewer #3: (No Response)

**Editorial and Data Presentation Modifications?**

Reviewer #2: Update the overview of Toxoplasma gondii strain/lineage diversity (2019–2024) by standardizing terminology (strain vs lineage vs genotype) and explicitly including non-archetypal lineages; consider adopting the following wording in the Introduction/Discussion: “Toxoplasma gondii shows global population structure that extends beyond the classical type I/II/III paradigm. Predominant clonal lineages in Europe and North America coexist with extensive genotype diversity and non-archetypal lineages elsewhere, which may have implications for virulence and disease severity.” Please also consider citing recent population/genomic reviews (e.g., Galal et al., 2019).

Present a standalone Ethics statement (committee name, protocol ID, guidelines) separate from methods text.

Clarify the in vivo timeline (infection → dosing initiation/frequency/duration → harvest) in one consolidated sentence; consider adding a small schematic.

Correct the Figure 3 caption to align with the data (PIM1 inhibits apoptosis).

In Statistics, specify biological n per group, normality checks, and any multiple comparison adjustments (or rationale if not applicable); consider non parametric tests where assumptions are not met.

Ensure figure resolution, consistent scale bars/labels, and uniform nomenclature throughout, temper overstatements in the Author Summary.

Please add a brief limitations paragraph clarifying that experiments address the acute tachyzoite phase only, that current standard therapy does not eradicate tissue cysts, and that the impact of PIM1 inhibition on chronic stages remains unknown

Reviewer #3: (No Response)

**Summary and General Comments**

Reviewer #2: Overall assessment and significance

The present version of the manuscript has improved in clarity and structure and addresses several prior comments. The central conclusion—that host PIM1 promotes Toxoplasma gondii intracellular proliferation by suppressing host-cell apoptosis, and that pharmacologic inhibition reduces parasite load in vivo—is supported by the data. The work is of interest for host directed therapy in toxoplasmosis and adds mechanistic insight by showing kinase activity dependence (kinase dead mutant and inhibitor).

Strengths

• Mechanistic coherence across complementary approaches (gain/loss of function, kinase dead PIM1, and in vivo inhibition).

• Consistence between in vitro and in vivo readouts regarding apoptosis and parasite burden.

• Potential translational relevance (host pathway targeting, combinability with current regimens).

Key weaknesses and remaining issues

• The overview of T. gondii strain/lineage diversity is outdated (over reliant on the I/II/III paradigm). PLOS requires current scholarship; please update 2019–2025, standardize terminology (strain vs lineage vs genotype), and explicitly include non archetypal lineages.

• The scope is limited to the acute post infection phase (tachyzoite stage). This should be stated explicitly in Abstract, Author Summary, Methods timeline, figure legends, and Limitations; implications for chronic/cyst stages should be framed cautiously.

• Statistical reporting needs tightening: specify biological n per group (in vivo), distinguish biological vs technical replicates, indicate normality checks for t tests, and clarify any multiple comparison control; consider non parametric tests where assumptions are uncertain.

• Minor presentation items: ensure a standalone Animal Ethics paragraph (committee, protocol ID, guidelines); clarify the in vivo timeline (infection → dosing → harvest) in one place; provide AZD1208 vehicle composition and a brief dose rationale. Also align any caption/text suggesting PIM1 “induces” apoptosis to reflect the data (PIM1 inhibits apoptosis).

Recommendation Minor Revision. No new experiments are required for this decision; the needed changes are editorial and reporting/interpretive clarifications that improve accuracy and transparency.

Reviewer #3: Authors have adequately addresses the concerns. The manuscript has been significantly improved than that of the previous versions. The manuscript can be accepted now. Congratulations to the authors.

PLOS authors have the option to publish the peer review history of their article (what does this mean? ). If published, this will include your full peer review and any attached files.

**Do you want your identity to be public for this peer review?** For information about this choice, including consent withdrawal, please see our Privacy Policy .

Reviewer #2: No

Reviewer #3: No

---

## [Editor Report · Acceptance letter]

Dear Lecturer Cao,

We are delighted to inform you that your manuscript, "Unraveling the role of host kinase PIM1 in *Toxoplasma gondii* infection: implications for therapies," has been formally accepted for publication in PLOS Neglected Tropical Diseases.

Best regards,

Shaden Kamhawi

co-Editor-in-Chief

Paul Brindley

co-Editor-in-Chief
